# Unveiling structural effects on the DC conductivity of warm dense matter via terahertz spectroscopy and ultrafast electron diffraction

Benjamin K. Ofori-Okai [1,2] ✉, Adrien Descamps [1,3,4], Edna R. Toro [1,5], Megan Ikeya [1], Stephanie B. Hansen [6], Mianzhen Mo [1], Andrew D. Baczewski [7], Danielle Brown[1,8], Luke B. Fletcher [1], Emma E. McBride [1,4], Xiaozhe Shen [1], Anthea Weinmann[1,9], Jie Yang [1], Jochen Schein[9], Zhijiang Chen [1], Xijie Wang [1,10,11] & Siegfried H. Glenzer [1] ✉

Understanding how materials under far-from-equilibrium conditions conduct electricity is vital for modeling planetary interiors, fusion energy, and other high-energy-density environments. Yet direct measurements of electrical conductivity in these states are challenging, as experiments must capture changes in both electronic conditions and atomic arrangement. Here we show, using laser-heated aluminum films, how the electrical conductivity of materials driven to the warm dense matter regime is influenced by temperature and structure. By directly measuring the electrical conductivity using terahertz time-domain spectroscopy and observing the atomic arrangement using mega-electron-volt ultrafast electron diffraction studies, we separate the impact of these different contributions on the observed sharp drop in the conductivity after laser heating. This approach is broadly applicable for measuring the electrical conductivity of matter laser heated to high-energy-density conditions. Our results are used to benchmark leading theoretical models and highlight the importance of accurately treating both electron and ion dynamics.

Many unresolved challenges in physics, from understanding the generation of planetary magnetic fields[1,2] to successfully driving a burning plasma for Inertial Fusion Energy[3,4], require a rigorous understanding of transport properties of matter in high-energy-density conditions. A critical parameter for simulating these processes is the zero-frequency (DC, or electrical) conductivity of materials in the warm dense matter

(WDM) regime[5–7]. Ideal plasma models, which assume the electron thermal energy, $E_k$, greatly exceeds the Fermi energy, $E_F$, exist for determining the DC conductivity[8,9]. In the WDM regime, $E_k$ is on the order of both $E_F$ and the mean potential energy, $\langle V \rangle$. This leads to a near unity degeneracy parameter, $\Theta = E_k/E_F \sim 1$, and Coulomb coupling, $\Gamma = \langle V \rangle/E_k \sim 1$. Consequently, ideal plasma theories break down as

[1]SLAC National Accelerator Laboratory, Menlo Park, CA, USA. [2]PULSE Institute, SLAC National Accelerator Laboratory, Menlo Park, CA, USA. [3]Aeronautics and Astronautics Department, Stanford University, Stanford, CA, USA. [4]School of Mathematics and Physics, Queen's University Belfast, Belfast, UK. [5]Mechanical Engineering Department, Stanford University, Stanford, CA, USA. [6]Pulsed Power Sciences Center, Sandia National Laboratories, Albuquerque, NM, USA. [7]Center for Computing Research, Sandia National Laboratories, Albuquerque, NM, USA. [8]Physics Department, Stanford University, Stanford, CA, USA. [9]Universität der Bundeswehr München, Neubiberg, Germany. [10]Faculty of Physics, University of Duisburg-Essen, Duisburg, Germany. [11]Department of Physics, TU Dortmund University, Dortmund, Germany. ✉e-mail: benofori@SLAC.stanford.edu; glenzer@SLAC.stanford.edu

quantum mechanical effects need to be considered, and the strong Coulomb coupling between ions leads to correlations reminiscent of crystal-like and liquid-like structures. Computational methods such as time-dependent density functional theory (TD-DFT), density function theory coupled with molecular dynamics (DFT-MD) using the Kubo-Greenwood formalism[10–17], or average atom density functional theory (DFT-AA)[18–20] have been developed to calculate the DC conductivity in WDM. These approaches require validation; thus, experiments that directly determine transport properties are essential.

Ultrafast laser pulses can drive thin metal films to WDM conditions[21]; however, as there is rapid heating of the electrons followed by much slower transfer of energy to ions, the resulting states are very far from equilibrium. The heated state is also highly transient, persisting for only picosecond durations[22–25]. This places strict requirements on the temporal resolution of any probe used to investigate laboratory-generated WDM. Furthermore, there is the challenge of developing accurate and consistent models that account for the electrons and ions being out of equilibrium. This always occurs on short timescales and is essential for comparison with experiments. Benchmark quality data must thus independently determine the atomic configuration, ion temperature, and electron temperature. Attempts have been made to estimate the DC conductivity from measurements of the high-frequency (AC, or optical) conductivity using femtosecond optical laser pulses[22,24,26–31], or by X-ray Thomson scattering measurements[12,32–34]. However, both approaches require models to infer the DC conductivity, and as such are susceptible to uncertainties in the models used. This highlights the need for direct measurement of the DC conductivity on ultrafast timescales.

Terahertz time-domain spectroscopy (THz-TDS) has been recently developed for studies of highly transient states, including WDM[35–39]. THz fields have such low frequencies that they effectively probe DC-like material properties. THz pulses can have durations on the order of picoseconds, allowing interrogation of highly transient phases. These properties make THz pulses an ideal probe of the conductivity of WDM. Similarly, mega-electron-volt ultrafast electron diffraction (MeV-UED) is well-suited for measuring the ionic structure of WDM[23,40–43] due to the high scattering cross section and electron mean free path comparable to the sample thickness required for laser excitation of thin films.

In this work, we combine THz-TDS with MeV-UED to interrogate the electrical conductivity of warm dense aluminum (WD-Al) driven by a short optical laser pulse. Our measurements use a custom table-top THz setup[44] and obtain data with excellent signal-to-noise with single-shot detection and direct measurement of the THz spectrum. We find that upon irradiation, there is a rapid monotonic decrease in the conductivity, with larger changes for increasing energy deposition. By combining THz and MeV-UED measurements, we disentangle changes in the conductivity due to increases in the electron scattering rate, which dominates on sub-ps timescales, from the concurrent atomic rearrangement as the sample transitions from solid to liquid, which takes place on the few-ps timescales. Finally, we use our data to assess state-of-the-art computational methods, finding that DFT-MD and DFT-AA calculations with the appropriate choice of exchange correlation functional and ionic structure match closely with our experimental data. This represents an important step towards providing benchmark-quality data to further improve theoretical models. Our results offer a direct measurement of the quantity of interest, the DC conductivity, near the static limit, while tracking the ionic structure. Consequently, the data offer constraints on calculations in the warm dense matter regime.

## Results

### Ultrafast electron diffraction

To investigate the evolution of ionic structure in the laser-generated WD-Al, UED measurements were performed at the SLAC MeV-UED facility[40,45]. The experimental setup is illustrated schematically in Fig. 1a. Free-standing 35 nm thick Al foils were irradiated by $\lambda = 800$ nm, 70 fs laser pulses with fluences, $F$, of up to 550 mJ cm$^{-2}$. The absorbed energy density, $\rho_E$, is defined as the amount of absorbed laser energy by the ambient material normalized to the mass of the material and is given by $\rho_E = FA/d_O\rho_M$, where $A = 13.5\%$ is the absorption ratio, $d_O$ is the film thickness, and $\rho_M$ is the mass density of Al. For the fluences used here, $A$ has been shown to remain unchanged[46]. The maximum $\rho_E$ reached was ~6.5 MJ kg$^{-1}$. Figure 1b–e show false-color diffraction patterns that are the average of 20 single-shot measurements at selected time delays, $\Delta t$, following irradiation with $199 \pm 20$ mJ cm$^{-2}$ fluence laser pulses. The corresponding absorbed energy density was 2.42 MJ kg$^{-1}$. Comparing $\Delta t = 0$ ps and $\Delta t = 2$ ps, there is a loss of higher-order rings in the diffraction patterns

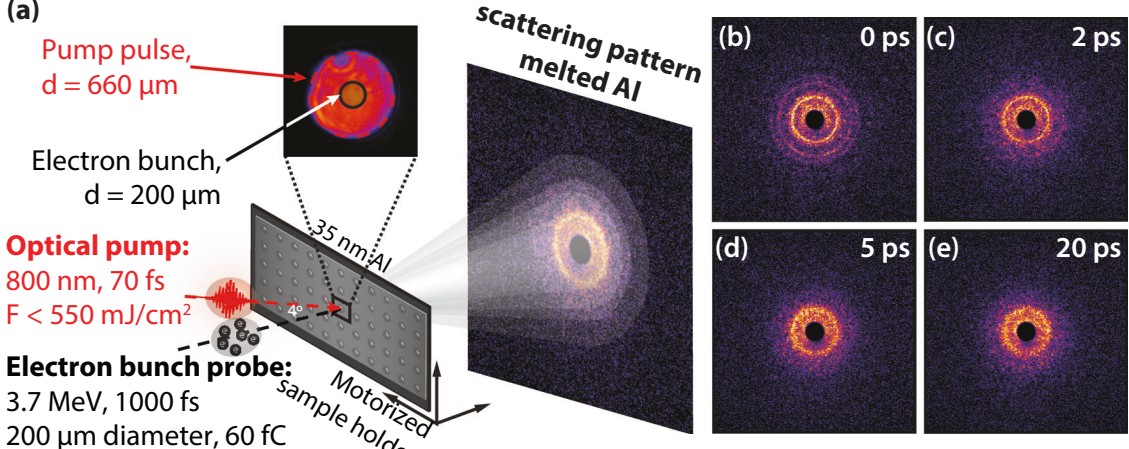

**Fig. 1 | Representative diffraction data collected using MeV-UED.** A schematic of the ultrafast electron diffraction setup is shown in (**a**) for studies of thin Al targets pumped by 70 fs, 800 nm laser pulses. The inset shows the near flat-top profile of the pump pulse, and the shaded region is the 200 μm region probed by the electron beam. The false color diffraction pattern is the average of 20 single-shot measurements. Images in (**b–e**) show false color images of the radial diffraction pattern at selected time delays. Each image is the average of 20 measurements. Samples were irradiated with a fluence of ~200 mJ cm$^{-2}$, corresponding to an absorbed energy density of 2.42 MJ kg$^{-1}$. The patterns show clearly evidence of solid material at up to 2 ps delay, but by 5 ps the sharp peaks indicative of solid material are replaced by broad rings.

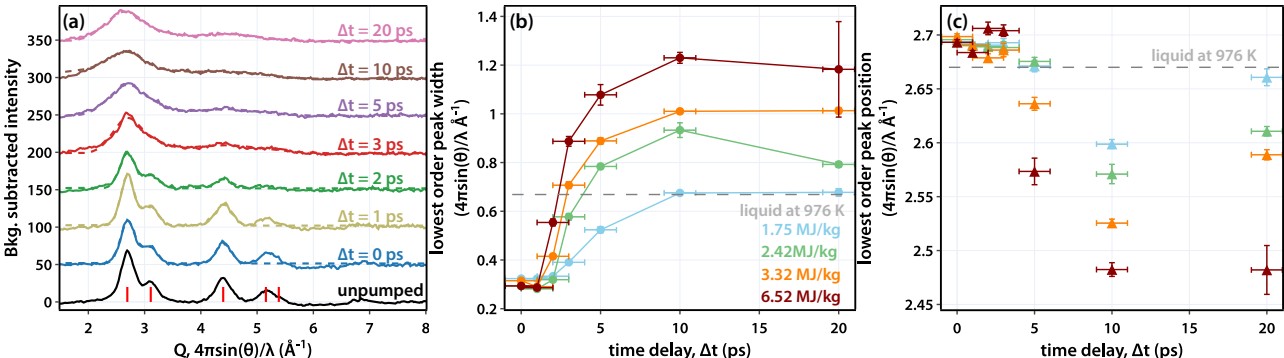

**Fig. 2 | Results of fits of diffraction data.** Plots in **a** show azimuthally integrated patterns at different time delays, along with one of an unpumped sample. Red lines correspond to expected diffraction peak positions for solid Al. By 5 ps there is no evidence of solid material, and higher order diffraction for $Q > 4\,\text{Å}^{-1}$ appears as a single broad feature. In **b** are plots of the FWHM of the lowest order peak. The gray dashed line corresponds to the width of the calculated $I(Q)$ based on neutron scattering measurements. In all cases, by $\Delta t = 5$ ps, the peak has broadened noticeably, indicating the disappearance of the solid. **c** shows the time evolution of the lowest order peak position in the diffraction pattern. Regardless of fluence, the lowest order peak remains at the same value at $\Delta t = 1$ ps, and only appreciably changes after $\Delta t = 5$ ps. Changes at 10 and 20 ps indicate continued evolution of the sample after melting. Error bars along the $\Delta t$ axis are the time-resolution of the instrument. Error bars along the $y$-axis are the RMS error of the fit.

consistent with the onset of melting. By $\Delta t = 20$ ps, only one broad feature is visible, suggestive of the formation of a liquid-like state.

Figure 2a shows radial lineouts plotted against momentum transfer $Q = 4\pi\sin(\theta)/\lambda_e$, with $\lambda_e \sim 0.34$ pm the de Broglie wavelength of the 3.7 MeV electrons, resulting from azimuthal integration of the measured diffraction patterns with background signals from inelastic and multiple scattering and dark current subtracted (see methods and supplementary text note 5). At $\Delta t = 2$ ps, the 111 Debye-Scherrer ring, located at $Q = 2.69 \pm 0.01\,\text{Å}^{-1}$, remains visible but appears broadened when compared with the diffraction pattern from an unperturbed sample. Additionally, there is a clear loss of intensity in the 220 ring ($Q = 4.40 \pm 0.01\,\text{Å}^{-1}$). After 5 ps, a single broad feature appears in a similar location as the 111 ring, and a weaker second broad feature can also be seen around the position of the initial 220 ring. These changes indicate loss of long-range order as the sample transitions from a crystalline solid into a more disordered liquid phase.

To quantify these changes, the radial lineouts were fit to either three Gaussians corresponding to the 111, 200, and 220 rings of the crystalline material or to two Gaussians for liquid diffraction. Parameters extracted from these fits are shown in Fig. 2b, c for different pump fluences and time delays. To follow the melting dynamics, we examined the full width at half maximum (FWHM) of the lowest order peak extracted from our Gaussian fits, shown in Fig. 2b. Of particular relevance to this study, in the liquid state, increased temperature leads to peak broadening[47–50]. We observe that the FWHM remains unchanged for $\Delta t < 2$ ps before increasing. The initial delay is attributed to the time required to sufficiently increase the lattice temperature to the melting temperature. After melting is complete, the width continues to change as the ions are heated. We determined when the sample has melted by comparing when the FHWM matches previously measured values. Based on analysis of neutron scattering experiments on liquid Al[50], we set a threshold FWHM, the dashed gray line in Fig. 2b, of $0.67 \pm 0.01\,\text{Å}^{-1}$ (see supplementary text note 6).

Details of changes in the lowest-order diffraction peak position are shown in Fig. 2c. For $\Delta t < 3$ ps, the lowest-order peak position remains essentially unchanged before decreasing and reaching an apparent minimum 10 ps after irradiation. The initial delay in the density change can be ascribed to an acoustic wave originating within a crystallite and traveling at the sound velocity. Given a longitudinal sound speed of $v_s = 6.4$ km s$^{-1}$[51], and assuming a crystallite size on the order of the film thickness, $d_O = 35$ nm, the time required to travel half the film thickness is $d_O/2v_s = 2.8$ ps, consistent with our observation. Equilibration then continues over longer time delays, indicated by the shift of the peak back to higher $Q$ at 20 ps delay at all but the highest irradiation.

Considering both the FWHM and lowest-order peak position, we infer that for $\Delta t < 2$ ps, our Al films retain a solid structure with near-solid density for all fluences of interest. At 3 ps delay, the Al film is largely composed of liquid-like material, and by 5 ps the sample has completely melted for energy densities exceeding 2.42 MJ kg$^{-1}$. For $\Delta t \geq 3$ ps, hydrodynamic expansion should be considered. These results are consistent with other recent MeV-UED measurements on Al[52], with our data confirming these observations over the energy densities relevant to our measurements. We use these details for interpreting changes in the THz data. Changes observed for $\Delta t = 1$ ps will thus primarily report on temperature effects, while after $\Delta t \sim 2$–3 ps will also be influenced by the modification of the ionic structure.

## Terahertz spectroscopy

To determine the conductivity, we performed single-shot THz spectroscopy measurements of 33 nm Al films heated to WDM conditions using intense femtosecond laser pulses[36,37,39,53]. Figure 3a shows a schematic of the experimental setup. The films were heated using $\lambda = 800$ nm pulses focused over a $\sim 2$ mm diameter area. To ensure probing of heated material, a thick sample card with holes <1 mm in diameter was used to mask the Al films. The highest fluence achieved was $\sim 330$ mJ cm$^{-2}$, and the corresponding $\rho_E$ was $\sim 5.1$ MJ kg$^{-1}$. The near single-cycle THz pulses transmitted through the sample were used to probe the conductivity. In our experiments, we varied the timing such that the THz pulse arrived 1 ps before the pump pulse ($\Delta t = -1$ ps) and up to 5 ps after the pump pulse ($\Delta t = 5$ ps).

Figure 3b shows example THz time-domain waveforms collected on unheated and laser-heated films at $\Delta t = 1$ ps. The THz time-domain waveforms were collected using an echelon-based electro-optic sampling method, which recorded the waveforms in a single shot[36,38,39,54,55]. Different colors correspond to different $\rho_E$, and measurements on an unheated film and a reference collected with no sample (the latter scaled by 0.1) are shown for comparison. Upon heating, the magnitude of the THz signal increases, $|E(t)|$, indicating an increase in the sample transmission. For the highest incident fluence, we observe an almost 4× increase in the transmission due to the changes in the electronic properties of Al ($|E(t)| \sim 0.16$ kV cm$^{-1}$ for the unheated film compared to almost 0.60 kV cm$^{-1}$).

To quantify the change in transmission, the frequency-domain spectra were calculated using a numerical Fourier transform of the time-domain data. When taking the Fourier transform, a hyper Gaussian window function, $\exp[-((t-t_0)/\tau)^{20}]$, was used[56] to remove contributions from signals outside the main cycles of the THz pulse; doing so limits the frequency resolution of the measurement to $\sim 0.6$ THz, with

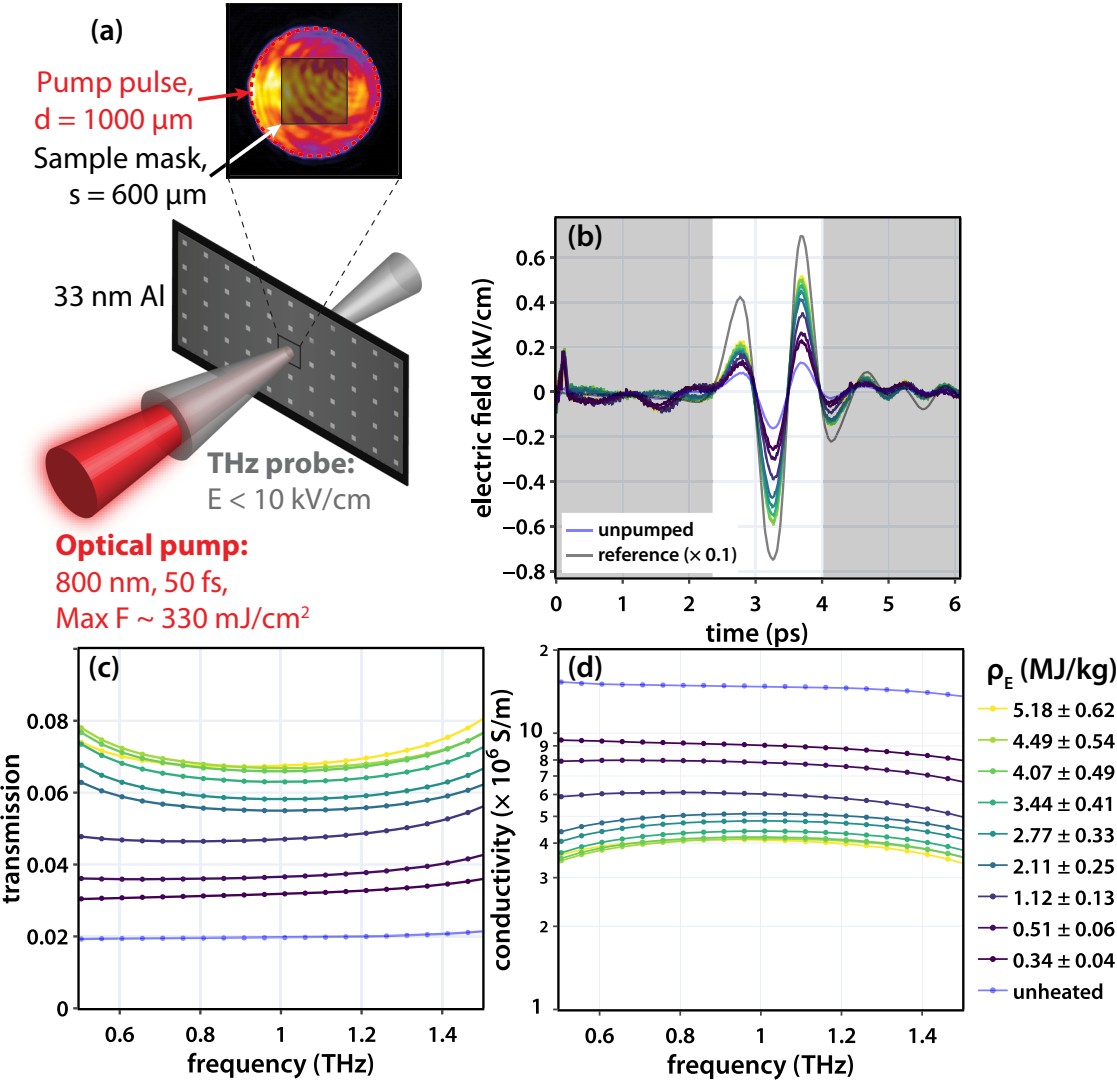

**Fig. 3 | Details of THz-TDS and representative data for pump-THz delay of 1 ps.** A schematic of the interaction point is shown in (**a**) and depicts the THz pulse transmission though thin Al targets pumped by 50 fs 800 nm laser pulses. The inset shows the pump pulse shape over a 1 mm diameter hole, and the shaded region marks the area unblocked by the sample card and thus probed by the THz pulse. THz time-domain waveforms are shown for an unheated Al film (blue), laser-heated Al film (red), and a reference with no sample (black). Plots in (**b**) show time-domain waveforms of the THz pulse passing through an unheated Al film (light blue), heated Al films (dark blue-yellow), and a reference (gray). The reference measurement is scaled by 0.1. The regions in gray were filtered out as described in the main text. In (**c**) are plots of the frequency-dependent transmission determined by normalizing the spectra of the unpumped and excited films to the reference spectrum. The relatively flat frequency dependence is indicative of Drude-like behavior. In (**d**) are plots of the frequency-domain conductivity obtained using the Tinkham formula. Consistent with the increase in transmission as the sample is excited, the average THz conductivity drops from $14.6 \pm 1.1 \times 10^6$ S m$^{-1}$ to $3.9 \pm 0.4 \times 10^6$ S m$^{-1}$.

additional points in the frequency domain arising from the length of the original time window. We normalized the spectra of the heated film, $\tilde{E}_s(\omega)$, to the spectrum taken with no sample, $\tilde{E}_r(\omega)$, and this was used to determine the frequency-dependent transmission, $t(\omega) = \tilde{E}_s(\omega)/\tilde{E}_r(\omega)$. The transmission magnitudes $|t(\omega)|$ are plotted in Fig. 3c. For the ambient film, the initial transmission is ~2%, and this increases to almost 7% when the film is excited to the highest energy density, consistent with the estimates derived by using only the peak of the THz electric field.

Figure 3d shows plots of the real part of the conductivity extracted via the Tinkham formula (see methods). We observe a largely frequency-independent conductivity over our THz bandwidth. For the ambient film, at 1 THz we determine a conductivity of $14.6 \pm 1.1 \times 10^6$ S m$^{-1}$. Independent four-point resistance measurements of the ambient film yield a conductivity of $17.4 \pm 3.5 \times 10^6$ S m$^{-1}$, closely matching our data and thus supporting the use of THz spectroscopy for probing WD-Al. We note that the lower conductivity is expected in thin

films, as there is increased electron scattering by grain boundaries and surfaces[57–60]; as such, care should be taken to ensure when extracting the conductivity of low-dimensional films. The increase in transmission as the sample is excited leads to a decrease in the conductivity, with values as low as $3.4 \times 10^6$ S m$^{-1}$ for the film excited to 5.1 MJ kg$^{-1}$.

Given the weak structure in the frequency dependence of the conductivity, we determined the average conductivity over the THz spectrum between 0.5 – 1.5 THz. The results of this averaging are shown in Fig. 4 for various energy densities and time delays. Beyond $\Delta t = 1$ ps, hydrodynamic expansion leads to a change in the film thickness, $d$, which is an input to the Tinkham formula. Frequency domain interferometry (FDI) measurements were performed to determine the change in the thickness which were then used to determine the conductivity (see supplementary text note 4). We observe a monotonic decrease in the conductivity with increasing energy density, consistent with faster electron scattering at higher electron temperatures. For a fixed energy density, there is an up to 2×

decrease in the conductivity between $\Delta t = 1\,\text{ps}$ and $\Delta t = 3\,\text{ps}$ and a much smaller decrease from $\Delta t = 3\,\text{ps}$ to $\Delta t = 5\,\text{ps}$. From the MeV-UED measurements, we expect that the ionic temperature of our Al films exceeds the melting temperature (~933 K, ~0.08 eV) within 5 ps. The conductivity trends observed here are expected to be due to changes in both the electronic and ionic temperatures. The former impacts the energy distribution of scattering electrons, whereas the latter impacts the electron-ion scattering cross sections.

## Discussion

The Drude model gives the frequency dependence of the conductivity as

$$\tilde{\sigma}(\omega) = \frac{n_e q_e^2 / m_e \nu_e}{1 - i\omega/\nu_e} \qquad (1)$$

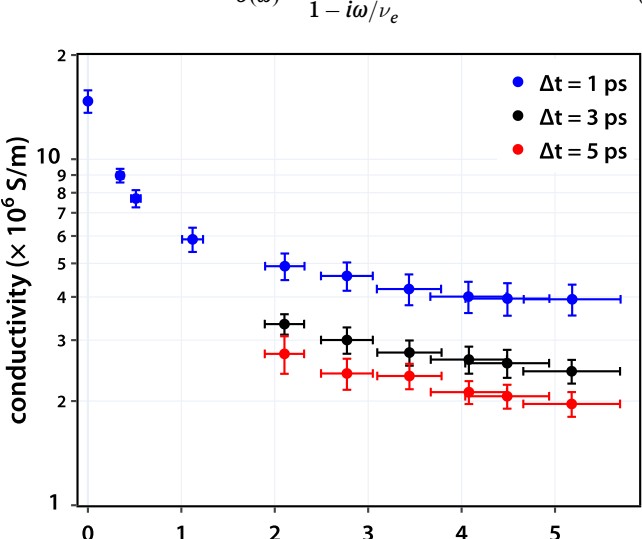

**Fig. 4 | Extracted average conductivity from THz measurements.** The average conductivity shows a consistent decrease with absorbed energy density within a given time delay. Each data point is determined from the average of three single-shot measurements. The error in the conductivity comes from the standard deviation when averaging across the THz spectrum. The error in the absorbed energy density is a combination of the variation in the sample thickness and the on-target laser energy.

where $n_e$ is the electron density, $q_e = 1.602 \times 10^{-19}\,\text{C}$ is the electron charge, $\nu_e$ is the total electron momentum scattering rate, and $m_e = 9.1 \times 10^{-31}\,\text{kg}$ is the electron mass. At THz frequencies, we expect $\omega/\nu_e \ll 1$; consequently, the conductivity is dominated by real part $\sigma_r = n_e q_e^2/(\nu_e m_e (1 + \omega^2/\nu_e^2)) \approx n_e q_e^2/\nu_e m_e = \sigma_0$, the DC value. Our data support this, as the imaginary part of the conductivity always remains small compared to the real part (see supplementary text note 2). Considering the melting time scales extracted from the UED data, we analyze the cases with $\Delta t = 1\,\text{ps}$ to focus on WD-Al with a solid structure and $\Delta t = 5\,\text{ps}$ for liquid-like WD-Al.

To determine the effects of temperature evolution and melting on the changes observed in the conductivity, we use Two-Temperature Model (TTM) calculations combined with details extracted from the UED measurements. The TTM was used to estimate the electron and ion temperatures, $T_e$ and $T_i$ respectively, following laser heating,

$$C_e(T_e)\frac{\partial T_e}{\partial t} = -G_{ei}(T_e)(T_e - T_i) + S(t) \qquad (2a)$$

$$C_i\frac{\partial T_i}{\partial t} = +G_{ei}(T_e)(T_e - T_i) \qquad (2b)$$

where $C_e(T_e)$ and $C_i$ are the electron and ion specific heat capacities, $G_{ei}(T_e)$ is the electron-temperature dependent electron-ion coupling parameter, and $S$ represents the energy deposition rate from the laser. $G_{ei}$ is a critical parameter of these simulations, which has been simulated with a variety of different models[61–64]. Details on other constants can be found in the methods section. To analyze the sensitivity of our predictions to this unknown parameter, we use values from Lin et al.[61] and Simoni et al.[62] which are among the smallest and largest values of $G_{ei}$ reported for density functional theory calculations relevant to our conditions.

Figure 5a shows the electron and ion temperatures for $\Delta t = 1\,\text{ps}$ (blue) and $\Delta t = 5\,\text{ps}$ (red) at various $\rho_E$. The bands thus represent the range of temperatures predicted between the $G_{ei}$ values under consideration. At our excitation conditions, $T_e$ never exceeds 1.2 eV, and given the high Fermi energy of Al, we do not expect ionization to change the electron density[12,14]. We thus ascribe the changes in conductivity to changes in the scattering rate. We observe that for both solid- and liquid-like WD-Al, the conductivity decreases with increased energy absorbed, corresponding to higher electron and ion temperatures, indicating more frequent electron scattering. $\nu_e$ is expected to depend on both $T_e$ and $T_i$, and this has been described according to $\nu_e(T_e, T_i) = \nu_{ee}(T_e) + \nu_{ei}(T_i)$ with $\nu_{ee}$ and $\nu_{ei}$ the electron-electron and

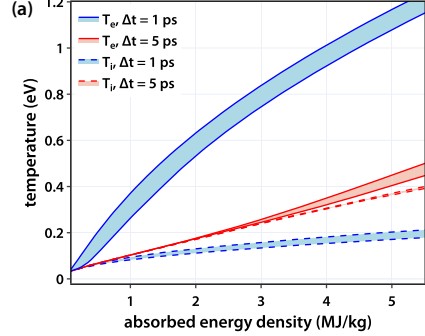

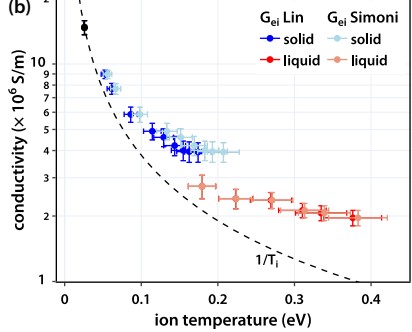

**Fig. 5 | Temperature calculations and ion-temperature dependence of the conductivity.** Curves in (**a**) show the electron and ion temperatures predicted by the TTM with different $G_{ei}$. The blue and red bands are the predictions for $\Delta t = 1\,\text{ps}$ and $\Delta t = 5\,\text{ps}$, respectively, which correspond to WD-Al with solid and liquid-like structures. **b** shows the dependence of electrical conductivity on $T_i$. The black dot is the room temperature value, and different shades of blue and red are used to distinguish between $T_i$ calculated with different $G_{ei}$. The dashed black line is the $1/T_i$

scaling predicted by condensed matter theory and is approximately observed at low temperature, but deviates significantly by the time $T_i$ is on the order of the melting temperature. The error in the conductivity comes from the standard deviation when averaging across the THz spectrum. The error in the ion temperature is a result of propagating the error in the energy density through the TTM calculations.

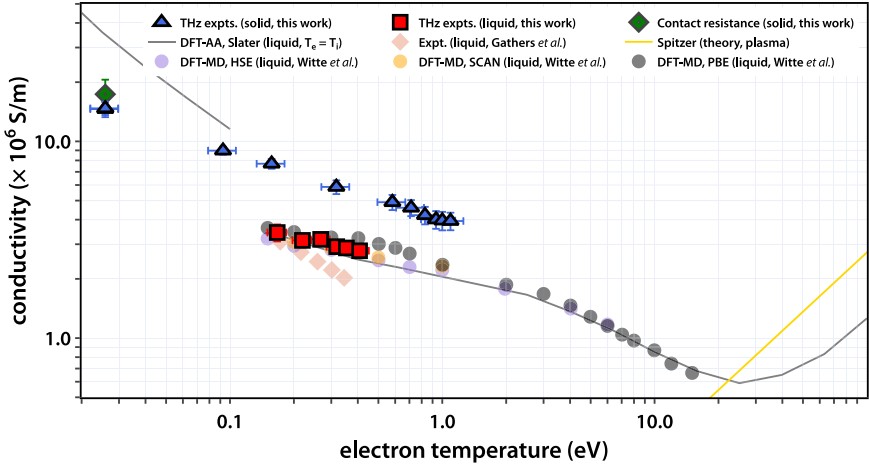

**Fig. 6 | Comparison of conductivity determined from THz measurements with past studies.** Blue triangles represent values extracted from THz measurements on Al with a solid structure and red squares are from THz measurements with a liquid structure. DFT-MD calculations by Witte et al. in purple, orange, and gray circles for HSE, SCAN, and PBE exchange correlation functionals, respectively, and DFT-AA calculations, shown as the gray line, use a liquid structure at solid density ($\rho_O = 2.7$ g cm$^{-3}$). For the THz measurements on liquids, the conductivity has been scaled to $\rho_O$. The Gathers' data are also scaled to solid density. The error in the conductivity comes from the standard deviation when averaging across the THz spectrum. The error in the electron temperature is a result of propagating the error in the energy density through the TTM calculations.

electron-ion scattering rates, respectively. For crystalline metals, $\nu_{ei} \propto T_i$ as the electron-phonon scattering rate depends on average phonon occupation, which is proportional to the ion temperature in these conditions. Fermi liquid theory predicts that $\nu_{ee}$ has a $T_e^2$ dependence; however, in our experiments, it remains much smaller than $\nu_{ei}$[65,66].

We test this in Fig. 5b, which shows the conductivity as a function of ion temperature inferred from the TTM using different $G_{ei}$ values[61,62], along with the $1/T_i$ contour indicating the expected conductivity due to the increased phonon occupation responsible for electron-ion scattering. For solid-structure WD-Al, the conductivity decreases with increasing ion temperature at a similar rate to the condensed matter theory predictions. For $T_i$ above 0.1 eV and especially for the long-time delay with liquid-like structure, the deviation from $1/T_i$ grows more significant, indicating that the electron-phonon coupling picture is no longer valid. Including a contribution from $\nu_{ee}$ would only further decrease the conductivity. Where the ion temperatures are similar, but the structures are different, the conductivity for the solid-structure WD-Al is higher than for the liquid-structure WD-Al. Similar behavior is observed near the melting point of Al, where the conductivity drops to ~40% of the solid value upon melting[67,68]. We observe a smaller change; our measured conductivity of liquid-like WD-Al is ~70% the solid-like WD-Al for the same ion temperature. The smaller difference is likely due to the elevated electron temperature for the solid-like WD-Al achieved in our experiments.

Ab initio computational methods have been employed to determine the electrical conductivity of WDM[11,20,69]. Previous calculations for WD-Al have utilized DFT-MD[14,70,71], DFT-AA[64,72,73], and NPA[13] methods. Importantly, there remain ongoing discrepancies between the results as approximations differ and central input parameters, most notably the exchange correlation functional, for different calculations are used[13,74]. Furthermore, recent experiments[12,33,34] only indirectly determine the DC conductivity, often relying on these same models. The data presented here give a direct measurement of the DC conductivity, providing model-independent values that offer critical tests for state-of-the-art computational tools.

Figure 6 shows a comparison of our THz conductivity measurements with experimental and theoretical results at different electron temperatures. For solid WD-Al, we present values from our experiments for $\Delta t = 1$ ps, and for liquid WD-Al, we show our data for $\Delta t = 5$ ps. For comparison with theory, we show recent results from DFT-MD simulations by Witte et al.[14]. Those conductivities were calculated using the Kubo-Greenwood formula within DFT calculations using three different exchange correlation functionals—the Perdew, Burke, Ernzerhof (PBE), the Heyd, Scuseria, Ernzerhof (HSE), and the Strongly Constrained and Appropriately Normed (SCAN) functionals. We also performed calculations based on a DFT-AA model[18] with a Slater exchange potential (3/2 LDA). All calculations were performed for a density $\rho_O = 2.7$ g cm$^{-3}$ and with $T_e = T_i$. Finally, plasma scaling expected from Spitzer theory is also plotted. In comparing these with THz experiments on liquid WD-Al, we focus on the $\Delta t = 5$ ps data, plotted as red squares, as the electrons and ions are expected to be in near thermal equilibrium; hence, these experimental conditions most closely match those of the simulations. As the calculations were performed for solid density, we scale the conductivity values extracted from the THz measurements, $\sigma_M$, by accounting for the change in sample thickness measured by FDI, $d = d_O + \Delta d$, according to $\sigma = \sigma_M (d_O + \Delta d)/d_O$. This accounts for the linear dependence of the conductivity on the electron density and the linear change in density expected due to the expansion of the thin film (see supplementary text note 11, Figure S12).

We draw several conclusions from these comparisons. The relationship between the conductivity and the ionic structure can be seen in the Ziman formula[18,20,75,76]:

$$\nu_{ei} = \frac{1}{3\pi Z_e} \int_0^\infty Q^3 S_{ii}(Q) \left| \frac{\partial \Sigma(Q)}{\partial \theta} \right| dQ \qquad (3)$$

where $\nu_{ei}$ is the electron-ion scattering rate, $Z_e$ is the number of free electrons per atom, $S_{ii}(Q)$ is the static ion-ion structure factor, and $\partial \Sigma(\theta)/\partial \theta$ represents the electron-ion collision cross section for electrons scattering at a given angle $\theta$. In this picture, increasing ion temperature in a crystalline solid leads to Debye-Waller broadening of sharp correlation peaks in $S_{ii}(Q)$, increasing electron-ion scattering. In general, the liquid phase will have larger electron-ion collision rates than the solid due to increased disorder, and those rates will be less sensitive to changes in the ion temperature. Importantly, liquid-metal models like the Ziman formula must be modified to exclude elastic scattering between identical lattice sites, which does not contribute[77,78]. For DFT-AA, we use structure factors from Baiko et al.[77] for the solid phase and self-consistent structure factors without correlation peaks for the liquid (see methods and supplementary text note 8).

From the electron-temperature dependence, we observe a decrease in conductivity for both the solid- and liquid-like WD-Al that is in qualitative agreement with predictions from DFT. For the solid-like WD-Al, our measured conductivities are ~3× higher than the measured conductivities in liquid-like WD-Al with similar $T_e$. This difference is largely driven by the change in structure factor upon melt, where a decrease by ~2.5× is expected from equilibrium measurements (and captured by the DFT-AA model). Importantly, comparison with values collected by Gathers et al. shows excellent agreement with our data in the liquid-like state. We also find good agreement between our liquid measurements, the DFT-AA calculations and the DFT-MD calculations using the HSE and SCAN functionals, with SCAN only slightly over-estimating the conductivity; a larger deviation is observed when the PBE functional is used. This is consistent with the HSE and SCAN functionals more accurately capturing the electronic structure of WD-Al (e.g., the L-shell orbital energies). All the DFT models show significant deviation from the Spitzer theory, which was developed for low-density plasmas.

These comparisons also highlight the importance of considering the details of how the electrons and ions deviate from thermal equilibrium. Such states are always generated during laser-solid interactions, and capturing such details remains a theoretical challenge. Our method provides a direct measurement of the conductivity and thus can validate models in- and out-of-equilibrium, thereby constraining wide-ranging multiphysics models. By augmenting our THz data with structure factors directly measured through UED experiments, we determine the electrical conductivity of matter in- and out-of-equilibrium and disentangle the contributions as the system evolves on ultrafast timescales.

In summary, we have demonstrated the use of a table-top setup for THz spectroscopy for studies of WDM and have shown that, in combination with MeV-UED measurements, we are able to extract quantitative information on the electrical conductivity of WD-Al in both the solid and liquid phase. We observe a drop in the electrical conductivity of WD-Al compared with ambient conditions and distinguish between changes in the conductivity based on changes to electron temperature and ionic structure. These measurements thus provide an important advancement towards obtaining benchmark quality data for verifying theoretical models, especially in the selection of structure factors and exchange-correlation models. While most previous approaches measure observables that require interpretation to obtain the transport property of interest, our approach measures the conductivity directly. This platform is also readily amenable to implementation with high-power lasers, which will enable access to higher temperature states and studies of materials beyond metals, such as inorganic oxide and hydrocarbons that are transparent and thus difficult to heat with through direct absorption. The method can also be combined with other diagnostics, such as XFELs[25,79–81], which would provide in situ characterization of density and electron and ion temperature. Such a measurement will provide unambiguous equation of state data of warm dense matter to constrain conductivity. Future work will explore these approaches and materials.

## Methods
### Sample preparation
Thin films of Al were made by two methods—evaporation onto a sodium chloride (NaCl) substrate or sputtering onto 20 nm of silicon nitride ($Si_3N_4$) on a silicon (Si) wafer. Evaporation onto NaCl substrates was performed at the Stanford Nano Shared Facility (SNSF). Sputtering onto $Si_3N_4$/Si wafers was performed by Silson Ltd., UK. Films evaporated onto NaCl were transferred onto a thick aluminum plate with 1 mm diameter holes, resulting in free-standing films. For films on $Si_3N_4$, the Si wafer was back etched to produce 600 μm x 600 μm square holes. Measurements on both films yielded consistent results.

### Ultrafast electron diffraction
The diffraction measurements were performed at the SLAC MeV-UED facility[40,45,82]. An amplified Ti:sapphire laser system capable of operating at 120 Hz supplied 12 mJ pulses used to generate the ultrafast electron bunches and the optical pump pulses. To produce the electron bunches, approximately 1 mJ of the Ti:sapphire output was frequency tripled and directed onto a copper photocathode; the resulting photoelectrons were accelerated to a kinetic energy of 3.2 MeV using a Linac Coherent Light Source–type photocathode radio frequency (rf) gun. The RF gun was powered by a pulse-forming network–based modulator and a 50-megawatt S-band klystron. The relativistic electrons were focused onto the target by two separate solenoids installed after the RF gun. 60 fC electron bunches were delivered in a 200 μm diameter spot at the sample, with a bunch length at the sample estimated to be 1 ps FWHM based on Debye-Waller measurements. Diffracted electrons were deposited onto a P43 phosphor screen located 3.2 m after the sample. A 1.6-mm-diameter hole was cut in the middle of the phosphor screen, and undiffracted electrons passed through this hole. An $f = 40$ mm focal length lens was used to image the phosphor screen onto an electron-multiplying charge-coupled device camera (Andor iXon Ultra 888).

The optical pump pulses were time-delayed using a mechanical delay stage, and a pinhole was utilized to select a uniform portion of the pump pulse profile with energy fluctuations <10% over the region probed by the electrons. A polarizer and a computer-controlled half-waveplate were used to modulate the pump pulse energy. The pump pulse was directed at ~10° to the sample normal.

For data collection, the laser was chopped to 30 Hz and computer-controlled shutters were used to select individual pulses. Each data point was the average of single-shot measurements on 20 distinct targets. For each target, undriven and driven diffraction patterns were collected and averaged offline.

### THz spectroscopy
THz measurements were performed using a home-built single-shot THz-TDS setup based on a previous experimental design[36,39]. The setup was driven by a Coherent Astrella Ti:sapphire laser system capable of producing 8.5 mJ, 50 fs pulses at a maximum repetition rate of 1 kHz. The THz and heating pulses were derived from the same laser source, and as such Δt could be set. The laser output was split, delivering up to 4.5 mJ pulses to heat the samples. Their energy was adjusted using a computer-controlled half-waveplate and a pair of thin-film polarizers. The pump pulses were then directed to a delay stage to set their arrival time relative to the THz pulses and focused using a 75 cm lens. Like the UED experiments, it was important to ensure uniform illumination. Consequently, the focal point of the lens was placed after the target plane. Using an upstream aperture to select a uniform portion of the pump beam would have significantly reduced incident fluence on the target, as the long THz wavelength prevents tight focusing.

4 mJ of laser energy was used for THz generation and detection. Of this energy, a beam splitter selected 3.5 mJ, which was used to produce THz pulses via optical rectification in a N-benzyl-2-methyl-4-nitroaniline (BNA) optically glued to a sapphire substrate[83,84]. The resulting THz pulses were directed using a set of 90° off-axis parabolic reflectors to the target plane, where they probed the sample. A pair of downstream off-axis parabolic reflectors was used to direct the transmitted pulses to a 2 mm thick (110)-cut zinc telluride (ZnTe) crystal. Holes in the off-axis parabolic reflectors and a downstream imaging system were used to check sample placement and overlap the THz and pump pulses at the target position.

THz detection was accomplished using the remaining 0.5 mJ pulses for echelon-based single-shot electro-optic sampling[35,36,38,39,54]. The sampling pulses were attenuated using a combination of a half-waveplate and polarizer before being directed to a second delay stage. The beam was expanded to overfill an 18 mm × 18 mm square echelon.

The echelon, produced by Sodick F.T., consisted of 120 steps with a width and height of 150 μm and 7.5 μm, respectively. Upon reflection off the echelon, a single sampling pulse was split into a series of time-delayed beamlets, which were focused into the 2 mm thick ZnTe crystal using an $f = 100$ mm focal length lens. An NA = 0.1 objective and $f = 75$ mm focal length lens were then used to image the echelon onto an Allied Vision Manta camera. A quarter waveplate and Wollaston prism were used in the single shot electro-optic sampling. An illustration of the experimental setup is shown in supplementary Fig. S1a.

For data collection, the Astrella was set to a 1 Hz repetition rate and computer-controlled shutters were used to block specific pulses. For each target, 5 frames were collected with the pump and THz pulses blocked, 5 with pump pulses blocked and THz pulses unblocked, 1 with the pump and THz pulses unblocked, and 5 with pump pulses blocked and THz pulses unblocked.

As the sampling pulse propagated through the Wollaston prism, each frame contained two images of the echelon, denoted with the superscripts $L$ and $R$. The relative brightness change $\frac{\Delta I^{L,R}}{I_0^{L,R}}$ was determined using images collected with the THz pulse present, $I^L$ and $I^R$, and images collected with the THz pulse absent, $I_0^L$ and $I_0^R$, according to $\frac{\Delta I^{L,R}}{I_0^{L,R}} = \left(\frac{I^{L,R}}{I_0^{L,R}} - 1\right)$. In these images, the information about the THz time-domain waveform is encoded along the horizontal axis, and so the time-domain waveform was extracted by averaging the $\frac{\Delta I^{L,R}}{I_0^{L,R}}$ images along the vertical axis of the image. Finally, the resulting signal $S(t)$ was determined by subtracting the waveform obtained using the right image from the corresponding waveform resulting from the left image, i.e. $S(t) = \frac{\Delta I^L}{I_0^L} - \frac{\Delta I^R}{I_0^R}$[36,39].

This procedure yielded THz time-domain waveforms transmitted through the unpumped film, the heated film, and a reference. Each data point was the average of three targets. The experiments were performed in a nitrogen atmosphere with a relative humidity of <5%. $\Delta t = 0$ was determined from optical-pump THz probe measurements on a Si wafer (see supplementary text note 1).

From the transmission, we used the Tinkham formula[85], which accounts for multiple reflections within a film whose thickness is much smaller than the wavelength and when the conductivity is large, to determine the conductivity, $\sigma(\omega)$:

$$\tilde{\sigma}(\omega) = \frac{n_s + 1}{Z_0 d}\left(\frac{1}{\tilde{t}(\omega)} - 1\right) \quad (4)$$

$n_s$ is the refractive index of the substrate, 2.7 for $Si_3N_4$[86], $Z_0 = 377\ \Omega$ is the impedance of free space, and $d = 33$ nm is the thickness of the film. For $\Delta t = 1$ ps, we do not expect significant expansion of the film, as the MeV-UED measurements show no change in density. For $\Delta t \geq 3$ ps, we accounted for hydrodynamic expansion using frequency domain interferometry measurements (see supplementary text note 4) to determine the film expansion[22,87,88].

## Two-temperature modeling

In the form of the TTM described in the main text, we assume uniform heating through the thickness of the film, as electrons initially heated by the laser within the skin depth travel ballistically through the film, rapidly thermalizing with cold electrons through collisions. Furthermore, given the large spot size of the beam, we do not expect lateral thermal diffusion on the ps timescales of this work.

At each timestep of the simulation, we calculated the electron specific heat capacity according to $C_e(T_e) = \gamma T_e$, with $\gamma = 91.2\ J\ m^{-3}\ K^{-2}$[61], while the ion specific heat capacity was held constant at $C_i = 2.5 \times 10^6\ J\ m^{-3}\ K^{-1}$, the constant volume heat capacity in the Dulong-Petit limit, consistent with the expected isochoric heating. Calculations

using the constant pressure heat capacity, which differed by ~5%, can be found in the supplementary text note 9 and Fig. S10. The electron-temperature dependent electron-ion coupling parameter, $G_{ei}(T_e)$, is taken from DFT calculations by Lin et al.[61] and Simoni et al.[62] with different values used for the solid and liquid. Details of the error in the temperature are given in supplementary note 7.

## Average atom DFT calculations

The DFT-AA/Slater conductivities given in Fig. 6 of the main paper were calculated using Bemuze, a semi-relativistic version of the average-atom code Purgatorio[89] modified to include self-consistent ion structure following the quantum Ornstein-Zernicke approach from Starrett and Saumon[90]. Given an element, mass density, electron temperature, and exchange-correlation potential, average-atom models find self-consistent orbital wavefunctions from the Schrödinger or Dirac equations, orbital occupations and chemical potentials based on Fermi statistics and charge conservation, and electron-ion potentials from the Poisson equation. These orbitals were used to generate T-matrix differential cross sections that can be integrated over the energy-dependent electronic density of states to produce $Q$-dependent scattering terms suitable for the Ziman formulation of conductivity[18].

## Data availability

The authors declare that the data supporting the findings of this study are available within the paper and its Supplementary Information file. All other relevant data supporting the findings of this study are available on request.

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

## Acknowledgments

The authors acknowledge useful discussions with D. Kraus and R. Redmer, and L. Seipp for help with the FDI measurements. SBH and ADB acknowledge helpful discussions with Alina Kononov. This work is supported by the following: by the U.S. Department of Energy (DOE) Office of Science, Fusion Energy Science under FWPs 100182 awarded to SHG, and 100866 awarded to SHG; by the U.S. DOE, Laboratory Directed Research and Development program at SLAC National Accelerator Laboratory under contract no. DE-AC02-76SF00515 awarded to LBF and as part of the Panofsky Fellowship awarded to BOO; by the UK Research & Innovation Future Leaders Fellowship (MR/W008211/1) awarded to EEM. This article has been co-authored by employees of National Technology & Engineering Solutions of Sandia, LLC under Contract No. DE-NA0003525 with the U.S. DOE. X.J.W. acknowledges support by the Deutsche Forschungsgemeinschaft (DFG, German Research Foundation) through the Collaborative Research Center (CRC) 1242 (project No. 278162697, project C01 Structural Dynamics in Impulsively Excited Nanostructures) & the funding by the DFG Germany's Excellence Strategy - EXC 2033 - 390677874 – RESOLV. Work carried out SLAC MeV-UED facility was supported by the DOE BES Accelerator and Detector program, the SLAC UED/UEM Initiative Program Development Fund. Part of this work was performed at the Stanford Nano Shared Facilities (SNSF) RRID:SCR_023230, supported by the National Science Foundation under award ECCS-2026822. The authors own all right, title and interest in and to the article and are solely responsible for its contents. The United States Government retains and the publisher, by accepting the article for publication, acknowledges that the United States Government retains a non-exclusive, paid-up, irrevocable, world-wide license to publish or reproduce the published form of this article or allow others to do so, for United States Government purposes. The DOE will provide public access to these results of federally sponsored research in accordance with the DOE Public Access Plan https://www.energy.gov/downloads/doe-public-access-plan.

## Author contributions

Conceptualization: B.K.O.O., Z.C., and S.H.G. Data collection: B.K.O.O., A.D., E.R.T., M.I., M.Z.M., D.B., X.S., A.W., J.Y., Z.C., X.J.W., and S.H.G. Analysis and discussion: B.K.O.O., A.D., E.R.T., M.I., M.Z.M., S.B.H., A.D.B., D.B., L.B.F., E.E.M., A.W., Z.C., and S.H.G. Theory and Simulation: S.B.H. and A.D.B. Writing—original draft: BKOO. Writing—review & editing: B.K.O.O., A.D., E.R.T., M.I., M.Z.M., S.B.H., A.B.D., D.B., L.B.F., E.E.M., X.S., A.W., J.Y., J.S., Z.C., X.J.W., and S.H.G.

## Competing interests

The authors declare no competing interests.
