## [Transparent Peer Review file · Nature Communications]

Unveiling Structural Effects on the DC Conductivity of Warm Dense Matter via Terahertz Spectroscopy and Ultrafast Electron Diffraction

Corresponding Author: Professor Benjamin Ofori-Okai

Version 0:

Reviewer comments:

Reviewer #1

(Remarks to the Author)

Review of manuscript "Unveiling Structural Effects on the DC Conductivity of Warm Dense Matter via Terahertz Spectroscopy and Ultrafast Electron Diffraction"
Submitted to Nature Comm.

In this manuscript the authors employed terahertz spectroscopy with mega electron volt ultrafast electron diffraction to obtain direct measurement of the DC conductivity on ultrafast timescales. The measurements remove the requirement for additional parameters for this. This is the main and important contribution in this manuscript. The authors focus on the electrical conductivity of warm dense aluminum for this.

Comments:

- 1) WDM could be explicitly defined on pg. 2.
- 2) The authors correctly point out that other derivations of conductivity approaches require models to infer the DC conductivity and claim that the method employed in the submitted manuscript is a direct measurement without this requirement. DFT-MD and DFT-AA calculations with the appropriate choice of exchange correlation functional and ionic structure match closely with the authors experimental data. Does this not imply that the match of experiment to model requires a theoretical method to evaluate the exchange correlation functionals and ionic structures?
- 3) The authors critical description of the experimental results with regard to phase transitions is clear and excellent. Figure captions are all clear and describe well the details.
- 4) The real component of the conductivity is correctly employed the Tinkham formula and the frequency dependence is obtained via a critical application of the Drude model.
- 5) Minor comments: Pg. 43, There is only a reference number 8 at the sentence ending on line 886 but no simple details as to what approximation was employed. An incomplete sentence beginning only with the word "There" on line 922 needs completion.

In summary, this manuscript clearly describes experimental results that improve on earlier work due to the reduction of parameters needed to obtain conductivity information. Although there remains a need for improvement of a few parameters that are actually pretty well accepted already, this work provides excellent new data that will definitely aid in the improvement of theoretical understanding of conductivity models.

Reviewer #2

(Remarks to the Author)

The manuscript contains experimental data for the ionic structure of warm dense liquid aluminium and the dc conductivity of the same state of matter. The experimental results are compared to a number of theoretical results and also to different experimental results. They cover temperatures up to about 10eV.

I have several issues with this manuscript:

1.) The electron diffraction measurements and the THz conductivity measurements seem to be two different setups/experiments that were done at different times and with different pump lasers. Is that correct? Are both systems so stable and comparable that shot-to-shot and day-to-day variations do not play a role? I assume the different thicknesses of the Al foil of 33 nm and 35 nm for the two setups do not matter even though hydrodynamic motion (expansion) needs to be taken into account?

2.) There is no temperature measurement. There is an estimate of possible electron and ion temperatures using a two-temperature model. Several of the input parameters seem not trustworthy. For instance, a quick check in the NIST database shows quite some variation of the heat capacity of solid aluminum with temperature but the authors state that they used a single value for all cases. The absorption of light in matter is a notoriously difficult problem, in particular in warm dense matter, so the manuscript should acknowledge this difficulty and at the very least give a citation for the simple formula (S10) and note in which limit it is valid.

3.) The comparison with DFT results opens up a number of questions as well. The citation given is Witte et al. (PoP 2018). In it, all conductivity calculations seem to be for equilibrium aluminum. It is however not clear to me, which results (from THz spectroscopy) in this manuscript stem from equilibrated runs (featuring a large Δt) or stem from non-equilibrium runs ($\Delta t < 2$ ps or so). In particular, since the main figure (Fig. 6) has the abscissa labeled as "electron temperature". Similar problems exist in Fig. 5, where the ion temperature dependence is shown. What is the respective electron temperature?

Reviewer #3

(Remarks to the Author)

The authors combine ultrafast electron diffraction and terahertz spectroscopy to monitor the evolution of electrical conductivity of an aluminum foil melted by an ultrashort laser pulse. The strong point here is the use of the ultrafast electron diffraction which unambiguously provides the evolution of the structure of the photoexcited material. It is thus possible to accurately correlate the retrieved terahertz conductivity with instantaneous structure of the aluminum.

The manuscript is comprehensible and well written; I appreciate the faithful and careful documentation of all results and methods as well as the scientifically sound analysis. Altogether, since the manuscript well advances the understanding of behavior of photoexcited materials at the shortest time scales, I recommend it for publication in Nature Communications.

The following technical issues are minor (they do not have the potential to compromise the conclusions):

* My major objection concerns the frequency resolution displayed in Figs. 3c,d. The employed time window indicated in Fig. 3b is ~ 1.6 ps; this means that the frequency resolution is $1/1.6$ ps = 600 GHz, i.e., there are exactly two experimental points in Figs. 3c,d. All other points are artificial since they appear as a result of zero-padding (or more precisely, almost zero): in turn, they represent just an interpolation of the "experimental" points and do not carry out any information supported by experiments. I do not think that using a longer time-window would change the conclusions on THz conductivity (and I'm aware that it would lead to artifacts due to the ultrafast evolution of the system, [J. Opt. Soc. Am. B 17, 327 (2000)]). However, absolutely smooth data from any THz system (the more from a single-shot system which is generally noisier; in fact, the noise is clearly seen in Fig. 3b) is a highly visible hint that some suspicious operations were done with the experimental data. Please reconsider the analysis or just include a sentence on what is the real realistic frequency resolution.

* Although THz conductivity is confirmed to be close to the DC conductivity in this particular system, one should be aware that charge localization may occur in certain systems [e.g. in thin gold layers, Phys. Rev. B 76, 125408 (2007) or in most semiconductor nanoparticles], i.e., THz frequencies may not be generally low enough to permit a direct connection to DC conductivity. A certain reservation should thus be expressed when discussing equality between THz and DC conductivity in the manuscript.

* When presenting THz conductivity, consider whether it would not be more relevant to show (and easier to discuss) the sheet conductivity instead – the sheet conductivity should be insensitive to plasma expansion.

Minor remarks related to the presentation (these are formal only and it makes no sense to publish them along with the rest of the report)

* Consider transfer of the equations governing the two-temperature model into the main text (or at least refer to the equations explicitly) – now it is really difficult to decode what the two-temperature model is about. It seems to me that the G_{ei} value is temperature-dependent: if this is the case, it would be helpful to indicate this dependence explicitly, like in the case of the thermal capacitances.

* Although excitation conditions are described in the methods section, it would be helpful if the authors presented their opinion to which extent the conditions are compatible for ultrafast electron diffraction and THz measurements and point out if there are some possible aspects in which they may differ.

* Fig. 3b-d: Please label the curves for individual powers unambiguously – the current continuous color scale (with obscure number like 4.58) permits a readout with ± 1 MJ/kg which translates to $>100\%$ error for the lowest fluences. I also expect that the curves with lowest transmission/highest conductivity corresponds to unexcited sample, but the corresponding color

implies 2 ± 1 MJ/kg – it would be good to clarify this.

* I have nothing against showing absolute THz field intensities in kV/cm, nevertheless, what matters is the relative change (text at lines 158 - 159 could be slightly rephrased in this respect) – this avoids the need to specify whether this is the field just after the sample, inside the sample or at some particular position in front of/inside the sensor.

Version 1:

Reviewer comments:

Reviewer #1

(Remarks to the Author)

The authors have succeeded in convincing this reviewer that they have provided direct measurement of the DC conductivity of warm dense aluminum on ultrafast timescales.

The data presented therefore show that it is now possible to obtain definitive benchmark data for theoretical methods.

In particular, the authors have provided clarification regarding what is required in order for a full theoretical understanding of their experimental data. This is now greatly helped by the authors' responses to all reviewers' questions regarding experimental details such as sample temperatures, thicknesses, heat capacity models employed, DFT calculation references. The comments by the experimental reviewers were mostly directed at technical issues that were clarified by the authors, thus making theoretical work much easier since less assumptions are now required.

In summary, this manuscript will encourage the theory community to take new and further looks at the properties of warm dense elements since it is important for a large number of subjects ranging from astrophysical to fusion physics.

Reviewer #2

(Remarks to the Author)

The authors have answered my questions satisfactorily. I do not have any more comments or questions. I can recommend publication.

Reviewer #3

(Remarks to the Author)

Great work!

Reviewer #1 (Remarks to the Author):

Review of manuscript “Unveiling Structural Effects on the DC Conductivity of Warm Dense Matter via Terahertz Spectroscopy and Ultrafast Electron Diffraction”
Submitted to Nature Comm.

In this manuscript the authors employed terahertz spectroscopy with mega electron volt ultrafast electron diffraction to obtain direct measurement of the DC conductivity on ultrafast timescales. The measurements remove the requirement for additional parameters for this. This is the main and important contribution in this manuscript. The authors focus on the electrical conductivity of warm dense aluminum for this.

Response: We thank the reviewer their review of our work and concise summary of the results. Below we respond to the comments.

Comments:

1) WDM could be explicitly defined on pg. 2.

Response: We thank the reviewer for suggestion. We have modified the main text to include the following description of WDM on page 2:

Ideal plasma models, which assume the electron thermal energy, E_k , greatly exceeds the Fermi energy, E_F , exist for determining the DC conductivity^{8,9}. In the WDM regime, E_k is on the order of both E_F and the mean potential energy $\langle V \rangle$. This leads to a near unity degeneracy parameter, $\Theta = E_k/E_F \sim 1$, and Coulomb coupling $\Gamma = \langle V \rangle/E_k \sim 1$. Consequently, ideal plasma theories break down as quantum mechanical effects need to be considered and the strong Coulomb coupling between ions leads to correlations reminiscent of crystal-like and liquid-like structure.

2) The authors correctly point out that other derivations of conductivity approaches require models to infer the DC conductivity and claim that the method employed in the submitted manuscript is a direct measurement without this requirement. DFT-MD and DFT-AA calculations with the appropriate choice of exchange correlation functional and ionic structure match closely with the authors experimental data. Does this not imply that the match of experiment to model requires a theoretical method to evaluate the exchange correlation functionals and ionic structures?

Response: We thank the referee for this observation. It is certainly true that all DFT-based models require some choice of exchange correlation potentials – there are a wide range of theoretically motivated potentials to choose from – and models that calculate the DC conductivity directly may require additional choices: for example, DFT-MD models require care with respect to finite-size effects and extrapolation of the dynamic Kubo-Greenwood response to the DC limit, and DFT-AA models require care with respect to the ion structure factor.

We believe that the present work addresses the reviewer's questions and is uniquely constraining for models, in two important ways:

1. First, the THz probe enables a measurement of the dynamic response very near the DC (static) limit. As such, we do not have to invoke a model to translate the measured observable (e.g. via implicit or explicit extrapolation of a dynamic conductivity) into the quantity of interest – instead we measure the quantity of interest directly with the THz probe.
2. Second, the UED measurements provide a clear constraint on the ionic structure, eliminating one degree of freedom necessary for calculating the DC conductivity.

In a very important sense, the interpretation of other experiments *are constrained by models*; our experimental can data *provide constraints* to theoretical predictions. To highlight this distinction, we are adding the following sentences to the manuscript:

In the introduction, last sentence of the last paragraph:

This represents an important step towards providing benchmark quality data to further improve theoretical models. **Our results offer a direct measurement of the quantity of interest, the DC conductivity, near the static limit, while tracking the ionic structure. Consequently, the data offer constraints on calculations in the warm dense matter regime.**

In the conclusion, middle of the last paragraph:

These measurements thus provide an important advance towards obtaining benchmark quality data for verifying theoretical models, especially in the selection of structure factors and exchange-correlation models. **While most previous approaches measure observables that require interpretation to obtain the transport property of interest, our approach measures the conductivity directly.**

3) The authors critical description of the experimental results with regard to phase transitions is clear and excellent. Figure captions are all clear and describe well the details.

Response: We deeply appreciate this positive assessment!

4) The real component of the conductivity is correctly employed the Tinkham formula and the frequency dependence is obtained via a critical application of the Drude model.

Response: We thank the reviewer for this positive endorsement of our work!

5) Minor comments: Pg. 43, There is only a reference number 8 at the sentence ending on

line 886 but no simple details as to what approximation was employed. An incomplete sentence beginning only with the word “There” on line 922 needs completion.

Response: We thank the reviewer for catching these errors. We have removed the “There” at the end of the caption and adjusted the text, page 41:

For solids, we follow Potekhin *et al.*⁸ and approximate the inelastic scattering in solids using Coulomb logarithms defined in terms of the inelastic component of the dynamic structure factor and the scattering potential.

In summary, this manuscript clearly describes experimental results that improve on earlier work due to the reduction of parameters needed to obtain conductivity information. Although there remains a need for improvement of a few parameters that are actually pretty well accepted already, this work provides excellent new data that will definitely aid in the improvement of theoretical understanding of conductivity models.

Response: We thank and appreciate the reviewer for offering this supportive appraisal of our work!

Reviewer #2 (Remarks to the Author):

The manuscript contains experimental data for the ionic structure of warm dense liquid aluminium and the dc conductivity of the same state of matter. The experimental results are compared to a number of theoretical results and also to different experimental results. They cover temperatures up to about 10eV.

Response: We thank the reviewer for taking their time to read and review the manuscript. Below we respond to the comments.

I have several issues with this manuscript:

1) The electron diffraction measurements and the THz conductivity measurements seem to be two different setups/experiments that were done at different times and with different pump lasers. Is that correct? Are both systems so stable and comparable that shot-to-shot and day-to-day variations do not play a role? I assume the different thicknesses of the Al foil of 33 nm and 35 nm for the two setups do not matter even though hydrodynamic motion (expansion) needs to be taken into account?

Response: We thank the reviewer for raising this point. This is correct – the experiments were performed on different setups. The laser systems used in both experiments are commercial systems with shot-to-shot energy fluctuations <1%. In addition, for each set of measurements calibrations of the laser energy on target were performed to ensure that the absorbed energy density could be accurately determined and any drift in the laser energy was compensated. The largest source of error here was due to the spatial inhomogeneity of the excitation laser, the details of which are described in the supporting materials.

Regarding the sample thickness, measurements provided in the supplementary information show an estimated variation of ± 4.7 nm, which is larger than the nominal difference of 33 nm vs 35 nm. As such any variation in this regard we believe is already considered. The thickness variation did contribute to the error in the absorbed energy density, which was calculated assuming different film thicknesses. However, this error was propagated as described in the supplementary information.

2) There is no temperature measurement. There is an estimate of possible electron and ion temperatures using a two-temperature model. Several of the input parameters seem not trustworthy. For instance, a quick check in the NIST database shows quite some variation of the heat capacity of solid aluminum with temperature but the authors state that they used a single value for all cases. The absorption of light in matter is a notoriously difficult problem, in particular in warm dense matter, so the manuscript should acknowledge this difficulty and at the very least give a citation for the simple formula (S10) and note in which limit it is valid.

Response: We thank the reviewer for raising this point. Indeed, the lack of a direct temperature measurement is something we are keenly aware of as an issue. We do mention briefly in the concluding remarks that the combination of the THz probing with XFEL diagnostics, such as inelastic X-ray scattering would provide a means to a direct temperature measurement, either X-ray Thomson scattering to study the electron temperature, or milli-electron-volt inelastic X-ray scattering to measure the ion temperature. We have added some references of possible ideas.

Regarding the choice of heat capacity, the value chosen is the constant volume specific heat capacity, C_v , in the Dulong-Petit limit. We chose this because the sample is isochorically heated, and so initially the temperature changes occur under constant volume conditions. Additionally, the ion temperature very quickly exceeds the Debye temperature. We also checked the results of the TTM calculations with the heat capacity on the NIST database, which is the constant pressure heat capacity, C_p , and using the fully temperature dependent C_v under the Debye model, $C_v(T)$, and find a 5% difference in the temperature. We have included plots of both cases in the supplementary materials as well as in this response for clarity.

Figure S10. Comparison of constant pressure and constant volume heat capacities on TTM calculations. Results of calculations of the electron temperature are shown in **a**; corresponding plots of the ion temperature are shown in **b**. Solid lines show the results of TTM calculations using the constant pressure heat capacity, and dashed lines are for the constant volume heat capacity.

We have added the following text to the supplementary information, page 42:

Comparison of effect of constant pressure vs constant volume heat capacities on TTM

Figure S10 shows the results of TTM calculations using the constant volume and constant pressure heat capacities. The constant pressure heat capacity is taken from the NIST database and is parametrized by the ion temperature using the Shomate equation:

$$C_p = A + BT_i + CT_i^2 + DT_i^3 + ET_i^{-2}. \quad (\text{S12})$$

The constants A , B , C , D , and E are specified for the solid and liquid phase. The constant volume heat capacity is investigated in two cases: 1) the value from the Dulong-Petit limit, C_v^0 , and 2) the temperature dependent C_v using the Debye model. In this case,

$$C_v(T_i) = 3C_v^0 \left(\frac{T}{T_D} \right)^3 \int_0^{T_D/T} \frac{x^4 e^x}{(e^x - 1)^2} dx. \quad (\text{S13})$$

Here $T_D = 428$ K is the Debye temperature of Al. As expected, the higher constant pressure heat capacity results in a lower final temperature of the system. The absolute difference is largest for the highest energy density simulated and corresponds to an $\sim 5\%$ difference.

The other values used in the TTM are drawn from computations and are widely used: the electron heat capacity is computed using density functional theory by Lin et al. [*Physical Review B* **77**, 075133 (2008)]. Notably, this value is also in good agreement with analytical models [Ashcroft, N. W. & Mermin, N. D. *Solid State Physics*. (Brooks/Cole, Belmont, CA, 1976)]. We also added the following text to the description of the Two-Temperature modeling in the methods section, page 18:

At each timestep of the simulation, we calculated the electron specific heat capacity according to $C_e(T_e) = \gamma T_e$, with $\gamma = 91.2$ J/(m³ K²)⁶¹, while the ion specific heat capacity was held constant at $C_i = 2.5 \times 10^6$ J/(m³ K), the constant volume heat capacity in the Dulong-Petit limit, consistent with the expected isochoric heating. Calculations using the constant pressure heat capacity, which differed by $\sim 5\%$, can be found in the supplementary materials. The electron-temperature dependent electron-ion coupling parameter, $G_{ei}(T_e)$, is taken from DFT calculations by Lin et al.⁶¹ and Simoni et al.⁶², with different values used for the solid and liquid

The formula in equation S10, which is also given in the main text of the manuscript, used to compute the absorbed energy density is defined as the amount of absorbed laser energy by the ambient material over the mass of the material. It can be converted to the form in the manuscript by considering:

$$\begin{aligned}
\rho_E &= \frac{\text{Absorbed energy}}{\text{mass}} = \frac{\text{Incident Energy} \times \text{Absorption Coefficient}}{\text{mass density} \times \text{sample volume}} \\
&= \frac{\text{Incident Fluence} \times \text{sample area} \times \text{Absorption Coefficient}}{\text{mass density} \times \text{sample area} \times \text{sample thickness}} \\
&= \frac{\text{Incident Fluence} \times \text{Absorption Coefficient}}{\text{mass density} \times \text{sample thickness}}
\end{aligned}$$

This is valid for our measurements and is true provided in initial density and the absorption are known. We agree that the absorption of light by WDM remains challenging but this complication is not relevant here since the absorption takes place before the sample has a chance to be driven to WDM conditions. As our fluences/intensities are relatively modest ($I < 10^{13}$ W/cm², $F < 600$ mJ/cm²), we assume a flat absorption constant. Work by Genieys et al. [Appl Phys A 126 263 (2020)] has shown the reflectivity of Al remains unchanged until about 1 J/cm² fluence, supporting this assumption. We have added to the sentence in the main text where the energy density is defined, and included the reference justifying the use of a constant absorption coefficient, page 4:

The absorbed energy density, ρ_E , is defined as the amount of absorbed laser energy by the ambient material normalized to the mass of the material and is given by $\rho_E = FA/d_0\rho_M$, where $A = 13.5\%$ is the absorption ratio, d_0 is the film thickness, and ρ_M is the mass density of Al. For the fluences used here, A has been shown to remain unchanged⁴⁶.

3) The comparison with DFT results opens up a number of questions as well. The citation given is Witte et al. (PoP 2018). In it, all conductivity calculations seem to be for equilibrium aluminum. It is however not clear to me, which results (from THz spectroscopy) in this manuscript stem from equilibrated runs (featuring a large delta t) or stem from non-equilibrium runs (delta t < 2ps or so). In particular, since the main figure (Fig. 6) has the abscissa labeled as "electron temperature". Similar problems exist in Fig. 5, where the ion temperature dependence is shown. What is the respective electron temperature?

Response: We thank the reviewer for raising this concern and apologize for the unintended ambiguity. The data in Figs. 5 and 6 are drawn from the same measurements and are plotted against different temperatures. For the data in Fig. 5, the corresponding electron temperatures are plotted in Fig. 6. In Fig. 6, for the data labeled as solid the electrons and ions are not in equilibrium; the liquid data are essentially in thermal equilibrium. The intended point of comparison is between the THz data at 5 ps and the theory curves, since these most closely match the theoretical conditions. We have modified the main text to highlight this, page 12:

“In evaluating the DFT simulations with THz experiments on liquid WD-Al, the intended focus is on the $\Delta t = 5$ ps data, plotted as red squares, as the electrons and ions are expected to be in near thermal equilibrium; hence, these experimental conditions most closely match those of the simulations.”

Below we use the results of TTM calculations to plot the data with the x- and y-axes are the ion and electron temperature, respectively:

The blue colored band is for the case where $\Delta t = 1$ ps, and is equivalent to using the values between the blue dotted lines shown in main figure 5 for the x-axis, and the blue solid lines shown in main figure 5 for the y-axis; the red band is the equivalent for $\Delta t = 5$ ps. The bands are bounded by different G_{ei} from Lin et al. [*Physical Review B* **77**, 075133 (2008)], and Simoini et al. [*Physical Review Letters* **122**, 205001 (2019)]. The dots on the plot are the T_e and T_i values where the measurements were made; the labels are the conductivities. We have added this figure to the supplementary materials as well.

Reviewer #3 (Remarks to the Author):

The authors combine ultrafast electron diffraction and terahertz spectroscopy to monitor the evolution of electrical conductivity of an aluminum foil melted by an ultrashort laser pulse. The strong point here is the use of the ultrafast electron diffraction which unambiguously provides the evolution of the structure of the photoexcited material. It is thus possible to accurately correlate the retrieved terahertz conductivity with instantaneous structure of the aluminum.

The manuscript is comprehensible and well written; I appreciate the faithful and careful documentation of all results and methods as well as the scientifically sound analysis. Altogether, since the manuscript well advances the understanding of behavior of photoexcited materials at the shortest time scales, I recommend it for publication in Nature Communications.

Response: We thank the reviewer for this positive endorsement and recommendation of our publication! Below we address the minor technical issues and remarks raised.

The following technical issues are minor (they do not have the potential to compromise the conclusions):

* My major objection concerns the frequency resolution displayed in Figs. 3c,d. The employed time window indicated in Fig. 3b is ~ 1.6 ps; this means that the frequency resolution is $1/1.6$ ps = 600 GHz, i.e., there are exactly two experimental points in Figs. 3c,d. All other points are artificial since they appear as a result of zero-padding (or more precisely, almost zero): in turn, they represent just an interpolation of the “experimental” points and do not carry out any information supported by experiments. I do not think that using a longer time-window would change the conclusions on THz conductivity (and I’m aware that it would lead to artifacts due to the ultrafast evolution of the system, [J. Opt. Soc. Am. B 17, 327 (2000)]). However, absolutely smooth data from any THz system (the more from a single-shot system which is generally noisier; in fact, the noise is clearly seen in Fig. 3b) is a highly visible hint that some suspicious operations were done with the experimental data. Please reconsider the analysis or just include a sentence on what is the real realistic frequency resolution.

Response: We thank the reviewer for raising this concern. We agree that applying the filter does reduce the frequency resolution of the method, and comply with the suggestion to add a sentence which states the frequency resolution of the measurement in the main text of the manuscript, page 7:

“When taking the Fourier transform, a hyper Gaussian window function, $\exp[-((t-t_0)/\tau)^{20}]$, was used⁵⁵ to remove contributions from signals outside the main cycles of the THz pulse; **doing so limits the frequency resolution of the measurement to ~0.6 THz, with additional points in the frequency domain arising from the length of the original time window.**”

* Although THz conductivity is confirmed to be close to the DC conductivity in this particular system, one should be aware that charge localization may occur in certain systems [e.g. in thin gold layers, Phys. Rev. B 76, 125408 (2007) or in most semiconductor nanoparticles], i.e., THz frequencies may not be generally low enough to permit a direct connection to DC conductivity. A certain reservation should thus be expressed when discussing equality between THz and DC conductivity in the manuscript.

Response: We thank the reviewer for raising this point. We are aware of this concern and furthermore we believe that this discrepancy remains relevant on short timescales if the sample is made up of grains or crystallites with transverse dimensions shorter than the electron mean free path. A detailed description of this goes beyond the scope of the current work, but we plan to investigate this in the future. We have strengthened our original reference to this issue in the manuscript when we discuss the conductivity values that we observed, page 8:

We note that the lower conductivity is expected in thin films, as there is increased electron scattering by grain boundaries and surfaces^{57–60}; **as such, care should be taken to ensure when extracting the conductivity of low-dimensional films.**

* When presenting THz conductivity, consider whether it would not be more relevant to show (and easier to discuss) the sheet conductivity instead – the sheet conductivity should be insensitive to plasma expansion.

Response: We thank the reviewer for this suggestion. While it is true that the sheet conductivity is insensitive to plasma expansion, the quantity is extrinsic which makes it difficult to compare with simulations which calculate the intrinsic conductivities. Additionally, most HED applications use the conductivity as an input; because of this and to avoid the complication of still having to normalize by the thickness to compare with the simulated values, we instead show the conductivity.

Minor remarks related to the presentation (these are formal only and it makes no sense to publish them along with the rest of the report)

* Consider transfer of the equations governing the two-temperature model into the main text (or at least refer to the equations explicitly) – now it is really difficult to decode what the two-temperature model is about. It seems to me that the G_{ei} value is temperature-dependent: if this is the case, it would be helpful to indicate this dependence explicitly, like in the case of the thermal capacitances.

Response: We thank the reviewer for this suggestion. We have moved the two-temperature model equations into the main text, along with the definitions for the various terms in the equations. Additionally, we have explicitly labeled the electron temperature dependence of G_{ei} . The equations now read:

$$C_e(T_e) \frac{\partial T_e}{\partial t} = -G_{ei}(T_e)(T_e - T_i) + S(t)$$
$$C_i \frac{\partial T_i}{\partial t} = G_{ei}(T_e)(T_e - T_i)$$

* Although excitation conditions are described in the methods section, it would be helpful if the authors presented their opinion to which extent the conditions are compatible for ultrafast electron diffraction and THz measurements and point out if there are some possible aspects in which they may differ.

Response: We thank the reviewer for this suggestion. We have added some discussion of this in the methods section:

Like the UED experiments, it was important to ensure uniform illumination. Consequently, the focal point of the lens was placed after the target plane. Using an upstream aperture to select a uniform portion of the pump beam would have significantly reduced incident fluence on target, as the long THz wavelength prevents tight focusing.

Both experiments used relatively modest fluences ($< 600 \text{ mJ/cm}^2$). As it is possible to focus the electron beam to $\sim 200 \text{ }\mu\text{m}$, it would be possible to achieve higher fluences for the UED measurements. Additionally, we expect better spatial for the UED measurements. Using an upstream aperture to select a uniform portion of the pump beam would significantly reduce achievable energy densities for THz measurements because of the large THz focal spot size.

* Fig. 3b-d: Please label the curves for individual powers unambiguously – the current continuous color scale (with obscure number like 4.58) permits a readout with $\pm \sim 1 \text{ MJ/kg}$ which translates to $>100\%$ error for the lowest fluences. I also expect that the curves with lowest transmission/highest conductivity corresponds to unexcited sample, but the corresponding color implies $2 \pm 1 \text{ MJ/kg}$ – it would be good to clarify this.

Response: We thank the reviewer for this suggestion. We have made this change, and added an extra label for the unheated film.

* I have nothing against showing absolute THz field intensities in kV/cm, nevertheless, what matters is the relative change (text at lines 158 - 159 could be slightly rephrased in this respect) – this avoids the need to specify whether this is the field just after the sample, inside the sample or at some particular position in front of/inside the sensor.

Response: We thank the reviewer for this suggestion. We have adjusted the sentence to shift the focus more to the sample transmission, with the THz field values as parentheticals to offer supporting information, page 7:

Upon heating, the magnitude of the THz signal increases, $|E(t)|$, indicating an increase the sample transmission. For the highest incident fluence, we observe an almost 4× increase in the transmission due to the changes in the electronic properties of Al ($|E(t)| \sim 0.16$ kV/cm for the unheated film compared to almost 0.60 kV/cm).

Reviewer #1 (Remarks to the Author):

The authors have succeeded in convincing this reviewer that they have provided direct measurement of the DC conductivity of warm dense aluminum on ultrafast timescales. The data presented therefore show that it is now possible to obtain definitive benchmark data for theoretical methods.

In particular, the authors have provided clarification regarding what is required in order for a full theoretical understanding of their experimental data. This is now greatly helped by the authors' responses to all reviewers' questions regarding experimental details such as sample temperatures, thicknesses, heat capacity models employed, DFT calculation references, The comments by the experimental reviewers were mostly directed at technical issues that were clarified by the authors, thus making theoretical work much easier since less assumptions are now required.

In summary, this manuscript will encourage the theory community to take new and further looks at the properties of warm dense elements since it is important for a large number of subjects ranging from astrophysical to fusion physics.

Response: We deeply appreciate this positive assessment and perspective of the impact of our work and thank the reviewer for recommending publication!

Reviewer #2 (Remarks to the Author):

The authors have answered my questions satisfactorily. I do not have any more comments or questions. I can recommend publication.

Response: We appreciate this positive review of our work and thank the reviewer for their recommendation!

Reviewer #3 (Remarks to the Author):

Great work!

Response: We appreciate the strong support for our manuscript, and thank the reviewer for their recommendation to publish!